# Carbon Biomass Estimation Using Vegetation Indices in Agriculture–Pasture Mosaics in the Brazilian Caatinga Dry Tropical Forest

**Vicente de Paula Sousa Júnior** [1], **Javier Sparacino** [2], **Giovana Mira de Espindola** [1,*]
**and Raimundo Jucier Sousa de Assis** [1]

1   Graduate Program in Development and Environment, Federal University of Piauí (UFPI),
    Teresina 64049-550, Brazil; vicentepsj@ufpi.edu.br (V.d.P.S.J.); raimundojucier@ufpi.edu.br (R.J.S.d.A.)
2   Instituto de Investigaciones Biológicas y Tecnológicas, Centro de Ecología y Recursos Naturales
    Renovables(CONICET), Universidad Nacional de Córdoba, Córdoba X5016GCA, Argentina;
    jsparacino@unc.edu.ar
*   Correspondence: giovanamira@ufpi.edu.br; Tel.: +55-86-99415-9208

**Abstract:** Remote sensing is valuable for estimating aboveground biomass (AGB) stocks. However, its application in agricultural and pasture areas is limited compared with forest areas. This study quantifies AGB in agriculture–pasture mosaics within Brazil's Campo Maior Complex (CMC). The methodology employs remote sensing cloud processing and utilizes an estimator to incorporate vegetation indices. The results reveal significant changes in biomass values among land use and land cover classes over the past ten years, with notable variations observed in forest plantation, pasture, sugar cane, and soybean areas. The estimated AGB values range from 0 to 20 Mg.ha$^{-1}$ (minimum), 53 to 419 Mg.ha$^{-1}$ (maximum), and 19 to 57 Mg.ha$^{-1}$ (mean). In Forest formation areas, AGB values range from approximately 0 to 278 Mg.ha$^{-1}$, with an average annual value of 56.44 Mg.ha$^{-1}$. This study provides valuable insights for rural landowners and government officials in managing the semiarid territory and environment. It aids in decision making regarding agricultural management, irrigation and fertilization practices, agricultural productivity, land use and land cover changes, biodiversity loss, soil degradation, conservation strategies, the identification of priority areas for environmental restoration, and the optimization of resource utilization.

**Keywords:** semiarid; aboveground biomass; remote sensing; Landsat

## 1. Introduction

The carbon cycle affects terrestrial ecosystems, including forests, family farming, and mechanized agriculture. Forests and family farming contribute to carbon sequestration, while mechanized agriculture can lead to greenhouse gas emissions [1–3]. While the mapping of carbon storage and sequestration has been extensively studied in forest regions [4–6], there is a lack of research focusing on semiarid areas [7–9] and on estimating carbon in different types of vegetation, including regions with agricultural activities [10].

In this sense, the carbon maintained by the aboveground living biomass is an ecological variable recognized as an Essential Climate Variable (ECV) by the Global Climate Observing System (GCOS) [11] due to its significant contribution to the global carbon cycle [12]. Biomass is crucial in carbon stock and essential to Reducing Emissions from Deforestation and Forest Degradation in Developing Countries (REDD+) projects [12]. Thus, understanding biomass in different land cover types is critical, especially in agricultural landscapes, where significant biomass can be found [13].

The agricultural sector in Brazil has a substantial impact on greenhouse gas emissions. According to the *Brazil Greenhouse Gas Emissions Report*, the agricultural sector contributed to 28.5% of emissions in the country, and when combined with changes in land use and

forest coverage, the emissions contribution reached 66.5% of all gas emissions [14]. Also, the Intergovernmental Panel on Climate Change [15] has emphasized significant opportunities for mitigation and adaptation in agriculture, forestry, land use, and oceans, which can be scaled up in the short term across most regions.

To address the negative impacts of greenhouse gas emissions, the Brazilian Government introduced the Low Carbon Agriculture Plan [16], which included measures to reduce emissions in the agricultural sector. The Integrated Systems (SI) approach, a vital plan proposal, combines crop production, livestock, and forestry on the same land, promoting soil fertility and biomass production [17]. Initiatives like this enhanced the need to assimilate remote-sensing data into crop monitoring and yield estimation models.

In 2005, a global-scale study revealed that croplands, pastures, and rangelands cover nearly half of the potentially vegetated land area on the planet [18]. In Brazil, although most agricultural activities are concentrated in the Cerrado biome [19–21], advances in technology and mechanization have enabled expansion into regions once considered unsuitable or unprofitable, such as the semiarid Caatinga region and the transitional areas between these biomes [22,23], such as the Campo Maior Complex (CMC) region.

Recent literature shows that climate change impacts vulnerable regions such as China [24,25]. In this sense, Brazil is no exception due to the diversity and complexity of its biomes, which are characterized by tropical forests, semiarid areas, and humid areas, among others. However, research on aboveground biomass (AGB) estimation in semiarid regions is limited compared with other sites, mainly due to the costs, time required for obtaining in situ data, and detail and periodicity of the data [26–28]. Few studies have specifically addressed AGB estimation in the Brazilian semiarid region across varying scales, including small, medium, and large areas, or agricultural sites [10,29,30].

In addition, firewood extraction, pasture expansion, and monoculture are prominent contributors to environmental degradation in the Brazilian Caatinga biome [14,15,23,31,32]. It is crucial to acknowledge that this degradation has transpired over the past 500 years [22,33], encompassing intricate complexities and fragility resulting from unsustainable land use practices. This combination of factors renders the Caatinga region highly susceptible to desertification while posing challenges to biodiversity conservation efforts [34,35].

Particularly in the CMC region, adopting commodity-focused agriculture strategies, such as soybean production, has caused significant degradation [31]. In this context, few studies in different areas have attempted to estimate carbon concentrations and to develop technical tools to assess the impact of agricultural practices and land use change on carbon biomass stocks, thus contributing to greenhouse gas emissions [7,10,36,37].

Regarding the estimation of aboveground biomass (AGB) by employing modified vegetation indices (MVIs), a study was conducted explicitly based on the Modified Soil-Adjusted Vegetation Index (MSAVI) [38]. Previously, another study indicated that using MVIs does not outperform the Normalized Differential Vegetation Index (NDVI) [39]. Another study found that the Soil Adjusted Vegetation Index (SAVI) performed better than the NDVI over lengthy periods in the Caatinga region [40]. Nevertheless, in arid and semiarid areas, the consistent use of NDVI is not recommended [41]. In this context, Nascimento [10] proposed the exploration of the lesser-utilized MVIs and Vegetation Indices (VIs) as an alternative to the widely adopted NDVI.

The use of remotely sensed data for the AGB estimation of agriculture–pasture sites in semiarid regions is helpful for inputs into crop growth models, crop management applications, and, crucially, for developing sustainable agricultural systems. In this context, the overall goal of this study is to investigate the potential of carbon biomass estimation (AGB) using Landsat 8 in agriculture–pasture mosaics located in a semiarid region of the Brazilian Caatinga biome. The specific objectives are (1) to evaluate the ability of Landsat 8 vegetation indices for AGB estimation in agriculture–pasture mosaics; (2) to assess the correlation between each pair of vegetation indices used; and (3) to test the potential of crop biomass estimation in different land cover types.

## 2. Materials and Methods

### 2.1. Study Area

The Brazilian Caatinga is located in the northeast of the country and covers the states of Bahia, Sergipe, Alagoas, Pernambuco, Paraíba, Rio Grande do Norte, Ceará, Maranho, Minas Gerais, and Piauí [32]. It has an area of 844,453 km$^2$ and accounts for 11% of the country's territory. Figure 1 shows the location area overlapped by the Kernel density of rural properties within the site.

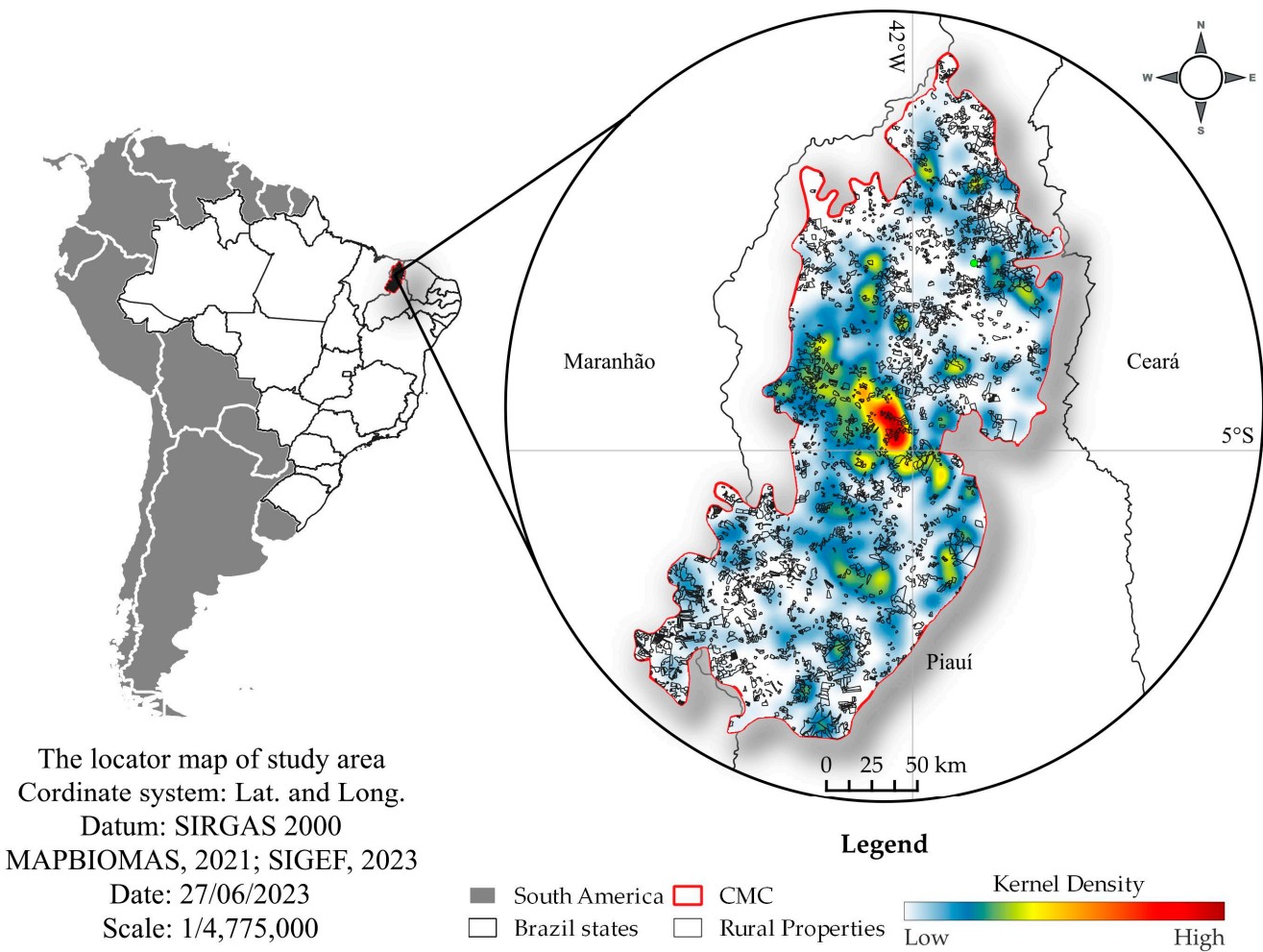

**Figure 1.** The location of the Campo Maior Complex (CMC) area is represented by the distribution of rural properties overlapped by its Kernel density. The rural properties are the sites of the agriculture–pasture mosaics for the year 2021.

With 41,420 km$^2$, the Campo Maior Complex (CMC) is nearly entirely in the Piauí state (76 municipalities) in Brazil, with a small portion in the Maranhão state (5 municipalities). More specifically, the CMC is located in the northwest of Caatinga, having 2,119,688 inhabitants, with 74% living in urban areas. The CMC contains the following conservation units (CUs): Sete Cidades National Park, Recanto da Serra Negra Natural Reserve, and Serra Ibiapaba Environmental Preservation Area [42–44]. The region has the perennial rivers Longá, Poti, and Parnaíba. The CMC is an ecotone between the Caatinga and joins the savanna-like Cerrado biome [30].

The CMC is located in a semiarid climate characterized by long dry spells and variable rainfall, primarily concentrated in a short rainy season and with an annual precipitation average of 1300 mm [30]. The region's climate is comparable to that of the Brazilian Caatinga. Due to the seasonality of rainfall, the vegetation adapts to periods of drought,

which makes studies and assessments of interannual rainfall variability essential [45]. In the CMC, herbaceous flora predominates, accentuating the physiognomy's similarity to the African savannah, and Carnaubas (native palms) may also be seen in the floodplains. Shrubby-tree vegetation is a characteristic of non-flooded areas [42].

The agriculture–pasture mosaics, represented by the rural properties within the site, encompass 301,109.80 hectares of the CMC's total land area of 4,120,000 hectares (Figure 1). It indicates that the studied area occupies roughly 7.31% of the CMC [46]. In 2021, the municipalities within the CMC generated 88.67% of the total silviculture production revenue of the Piauí state [47]. The production of grains, such as soybean and corn, is significant in the CMC, with soybean accounting for 2.62% and corn for 2.67% of the total agriculture production in Piauí. Regarding the occupied area, soybean production corresponds to 21,118.00 hectares, while corn production occupies 64,880.00 hectares [48]. Livestock activities requiring pasture, such as cattle, goats, sheep, and horses, respectively occupy 33.74%, 39.46%, 22.78%, and 34.75% of the total state's livestock. Notably, 99.27% of the buffalo herd in Piauí is located in the CMC region [49].

According to the data provided in the 2022 by the Annual Report on Deforestation in Brazil (RAD-2022) [50], 27,689.83 hectares of Caatinga forest were areas of deforestation in the CMC territory. This corresponded to 1613 deforestation alerts; of these, 1507 alerts were recorded in the Piauí portion of the CMC area while 106 alerts were in Maranhão state. Among the different factors contributing to deforestation, 2 alerts were identified as being related to illegal mining, 24 were linked to urban expansion, and 94 were associated with other forms of deforestation. Most cases of deforestation cases were attributed to agricultural activities. The smallest deforested area reported was 0.2 hectares, observed in the municipality of Alto Longá in the state of Piauí, and the largest deforested area measured 1275.11 hectares and was identified in the municipality of Cocal de Telha [50], also located in Piauí state.

### 2.2. MapBiomas Secondary Data

Secondary data was obtained from the Brazilian Annual Land Use and Land Cover Mapping Project (MapBiomas) [46], and this data was used to delimit the area of interest for modeling the aboveground biomass estimation. MapBiomas data are available for free on the MapBiomas platform and is an outcome of the cloud computing processing of Landsat series data for Brazil. The following land use and land cover (LULC) classes for the year 2021 were considered for delimitation in the analysis: Forest plantation; Pasture; Sugar Cane; Mosaic of uses; Soybean; Other temporary crops; and Forest formation. Only areas larger than three hectares were selected, and an influence area of 30 m was applied for further analysis.

### 2.3. Carbon Biomass Estimation Using Google Earth Engine

A flowchart of the steps performed in this study is described in Figure 2. Cloud computing processing on the Google Earth Engine (GEE) platform was performed to analyze the CMC's territorial area. Data was collected from Landsat 8, Level 2, Collection 2, Tier 1 satellite imagery, which has a spatial resolution of 30 m. The specific bands used in the processing were B2 (Blue), B3 (Green), B4 (Red), and B5 (Nir). The Tier 1 Level 2 data represents a processed and calibrated version of the satellite imagery, ensuring high quality and consistency for various applications.

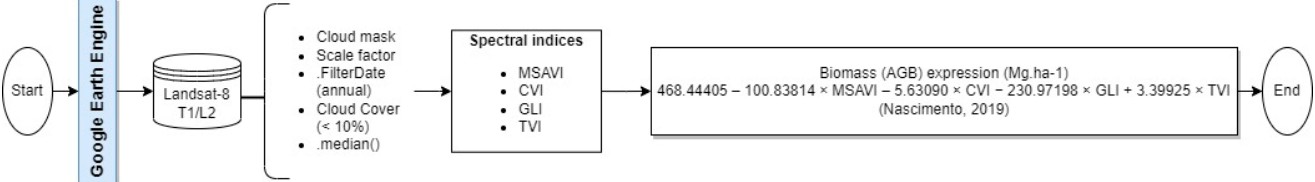

**Figure 2.** Flowchart of the JavaScript code implementation in Google Earth Engine.

Using GEE with data from the Landsat 8 collections enhanced the research purpose of open access. Landsat 8 OLI data was utilized for the period ranging from 2013 to 2022 to select all the imagery from this collection, as Landsat 8 was launched in 2013. The data underwent annual refinement processes, including applying cloud-masking techniques, scale factor adjustment, cloud-cover assessment, and collection reduction using the median function. Following that, the spectral indices described in Table 1 were computed.

**Table 1.** Description of name and definition of the vegetation indices used in this study.

| Vegetation Index (VI) | Equation |
|---|---|
| MSAVI<br>Modified Soil Adjusted Vegetation Index | $\frac{2(Nir+1)-\sqrt{(2Nir+1)2-8(Nir-Red)}}{2}$ |
| CVI<br>Chlorophyll Vegetation Index | $\frac{(Nir*Red)}{Green2}$ |
| GLI<br>Green Leaf Index | $\frac{(2\times Green-Red-Blue)}{(2\times Green+Red+Blue)}$ |
| TVI<br>Triangular Vegetation Index | $0.5\,[120\,(Nir - Green) - 2.5\,(Red - Green)]$ |

The estimation of the AGB was performed following the equation developed by Nascimento [10] for semiarid regions of the Brazilian northeast as follows:

$$468.44405 - 100.83814 \times \text{MSAVI} - 5.63090 \times \text{CVI} - 230.97198 \times \text{GLI} + 3.39925 \times \text{TVI}. \tag{1}$$

Initially, the estimator was applied to land use areas of Pasture classes and a generic class of Agriculture–pasture mosaics [10]. In this study, we included well-defined areas of Forest plantation, Pasture, Sugar Cane, Mosaic of uses, Soybean, Other temporary crops, and Forest formation. This inclusion is justified because the Pasture area constitutes most of the agriculture–pasture mosaics in the CMC. Additionally, it is essential to assess the application of Vegetation Indices (VIs) to semiarid regions.

Equation (1) is built from well-known vegetation indices. Each of them has unique features appropriate for the AGB estimation. The MSAVI can enhance the dynamic range of vegetation signals and reduce the influence of soil background, making it suitable for vegetation detection in semiarid areas [51]. The CVI minimizes variations in the leaf area index and exhibits greater sensitivity to leaf chlorophyll content [52]. The GLI utilizes visible bands to detect green leaves and stems, yielding improved results in areas without grazing [53]. Lastly, the TVI is based on the concept that chlorophyll absorption and near-infrared reflectance both lead to decreased red-band reflectance and increased total triangle area [54].

The annual temporal window was selected as an entire year to avoid misleading results associated with the seasonality of agricultural production that may occur within the Mosaic of uses and Other temporary crops classes. Therefore, the defined time frame encompasses the entire year rather than just the growth period of a specific category.

### 2.4. Correlation and Statistic Classes

Using data from MODIS (MYD11A1 V6.1) to estimate Land Surface Temperature (LST) and CHIRPS DAILY (Climate Hazards Group Infrared Precipitation with Station data) to estimate precipitation on the Google Earth Engine (GEE) platform, we created a chart that allows the comparison of these variables, LST, and precipitation with the correlated results between AGB values and VIs within the CMC area.

Correlation takes values between −1 and 1. Values close to 0 indicate a lack of correlation, while values close to 1 indicate a strong correlation and those close to −1 indicate a strong anti-correlation. Correlations between the AGB values were calculated using Equation (1), and each one of the VIs were obtained using the r.covar package that generates a correlation matrix between raster files [55]. Even though, by definition, the AGB results correlate with the VIs, it is relevant to understand the degree of each correlation

to identify whether one dominates the AGB dynamics and exhibits a stronger correlation. Correlations were also calculated between each pair of the VIs, and the results are presented as correlation matrices that characterize our region of interest from the years 2013 to 2022. The nondiagonal entries in the correlation matrix represent the correlation between two different features, introducing a standard way of merging various datasets [56].

The minimum, maximum, and average functions for each LULC thematic class (Forest plantation; Pasture; Sugar Cane; Mosaic of uses; Soybean; Other temporary crops) within the selected sites were obtained using zonal statistics. In our study, we used the Forest formation class for validation. The Forest formation class is the land cover type represented by natural vegetation, used here as a reference only. In spatial analysis, zonal statistics are used to calculate the statistics on values of raster data within the zones of another vector dataset. In our study, the raster data are the AGB estimations and the rural properties in each LULC thematic class represent the vector dataset. Then, the AGB values ($Mg.ha^{-1}$) were extracted, and the primary statistics function [55] was used by each LULC class, recording the minimum, maximum, and average values for each site. The sum values of the estimated AGB ($Mg.ha^{-1}$) per year and for each LULC class returned by the zonal statistics algorithm were also obtained. The final step was creating the maps in a printable layout to present the results.

## 3. Results

### 3.1. Biomass Estimation

Figure 3 shows the ten maps from 2013 to 2022 examined in this study. A gray hue was used as the background of the CMC territory to improve contrast. The results obtained can be consulted in the script developed in Google Earth Engine, which is freely accessible (GEE script available at: https://code.earthengine.google.com/f4aee75df401c3 2613cd5f64c1b2a177?noload=true on 8 August 2023) and (GEE script available at: https://code.earthengine.google.com/21474f124b0ea6b12bf1ed949763128c on 8 August 2023).

In Figure 4, we compared the first (2013) and the last year (2022) to show the AGB dynamics over the decade. We zoom in on aleatory regions to improve the visualization of the results. It is worth mentioning that subfigures are on different scales, and this approach was made to enhance data visualization only.

In Figure 4a, corresponding to the Forest plantation class, values have ranged significantly over the decade. While most of the area showed an increase in the AGB values, with the evident emergence of values greater than 60 $Mg.ha^{-1}$ in 2022, a small region in the western part presented a decrease in the AGB value. Other significant changes that occurred are presented in Figure 4c, where the Sugarcane class is shown. In 2013, the Sugarcane class provided values ranging from 15 to 30 $Mg.ha^{-1}$. However, in 2022, the situation is very different. While some patterns of 2013 remain, this class is no longer dominant, with values ranging from 30 to 45 $Mg.ha^{-1}$. In Figure 4b (Pasture), Figure 4d (Mosaic of uses), and Figure 4e (Soybean classes), there were changes in the estimated values of AGB, which dropped from 30–45 $Mg.ha^{-1}$ to 15–30 $Mg.ha^{-1}$ in specific sites of these classes. Figure 4f (Other temporary crops) showed an increase in AGB values from 15–30 $Mg.ha^{-1}$ to 30–45 $Mg.ha^{-1}$.

The LULC classes Pasture (Figure 4b), Mosaic of uses (Figure 4d), and Soybean (Figure 4e) all behaved similarly, lowering their biomass values from the 15 to 30 $Mg.ha^{-1}$ range. Also, it is crucial to note that because they are poorly defined land cover, the Mosaic of uses (Figure 4d) and Other temporary crops (Figure 4f) may exhibit diverse fluctuations. Another point to note is the region around the Soybean LULC class, which shows a considerable shift on the maps (Figure 4e) toward higher biomass values. Within the CMC region, crops are usually farmed near eucalyptus, which might justify this as a Forest plantation land cover type.

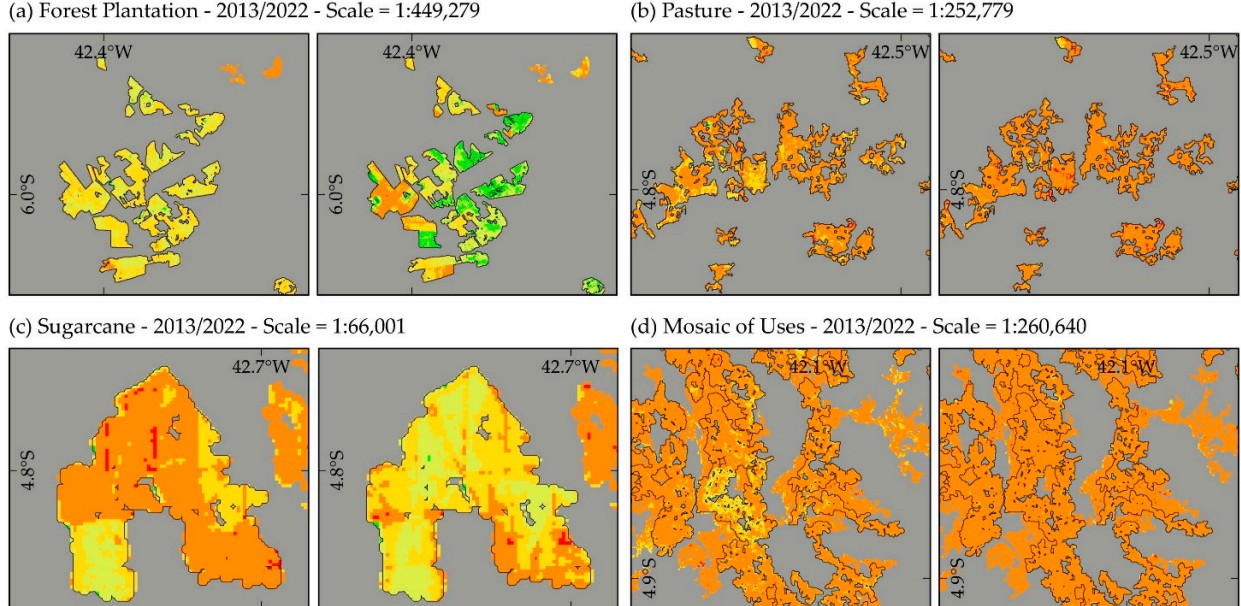

**Figure 3.** Estimated AGB (Mg.ha$^{-1}$) data for the CMC region from 2013 to 2022. The small rectangles delimit the sites presented in Figure 4.

(a) Forest Plantation - 2013/2022 - Scale = 1:449,279

(b) Pasture - 2013/2022 - Scale = 1:252,779

(c) Sugarcane - 2013/2022 - Scale = 1:66,001

(d) Mosaic of Uses - 2013/2022 - Scale = 1:260,640

**Figure 4.** *Cont.*

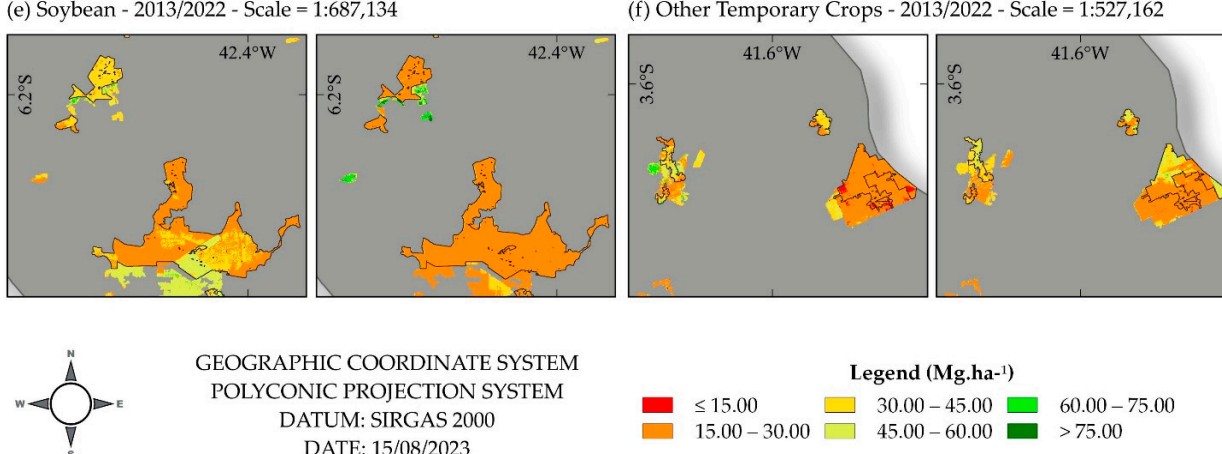

(e) Soybean - 2013/2022 - Scale = 1:687,134      (f) Other Temporary Crops - 2013/2022 - Scale = 1:527,162

GEOGRAPHIC COORDINATE SYSTEM
POLYCONIC PROJECTION SYSTEM
DATUM: SIRGAS 2000
DATE: 15/08/2023

**Legend (Mg.ha⁻¹)**

| | |
|---|---|
| ≤ 15.00 | 30.00 – 45.00 | 60.00 – 75.00 |
| 15.00 – 30.00 | 45.00 – 60.00 | > 75.00 |

**Figure 4.** Zoomed in view of LULC classes: (**a**) Forest plantation; (**b**) Pasture; (**c**) Sugarcane; (**d**) Mosaic of uses; (**e**) Soybean; (**f**) Other temporary crops, showing the results of estimated AGB (Mg.ha$^{-1}$) in 2013 and 2022.

### 3.2. Correlation between Estimated AGB and Vegetation Indices

Figure 5 shows the data of the remotely sensed estimations for the variables of precipitation and temperature from 2013 to 2022 in the CMC region. Since 2016, the estimated temperature has decreased, which may be affected by climatic conditions and data availability. Regarding precipitation, 2016, 2013, and 2015 correspondingly had the lowest cumulative annual averages. The temperature dropped as the average total yearly precipitation rose.

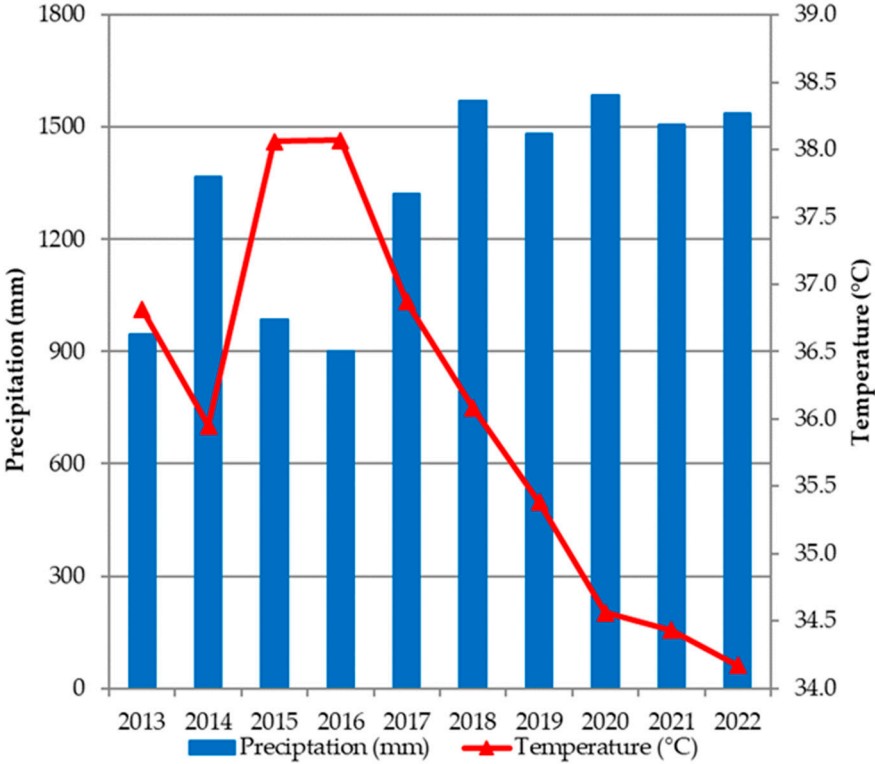

**Figure 5.** Estimated Land Surface Temperature (LST_Day_1km) and precipitation variables in the Campo Maior Complex region from 2013 to 2022.

Figure 6 illustrates the correlation matrix for each year. We included in Figure 6 the number of Landsat 8 images used to calculate the vegetation indices and the estimated biomass. The year with more Landsat 8 images available was 2016, with 49 images, while

the year with less was 2021, with 31 images. Regarding the correlation between the variables in the correlation matrix, 2018 and 2022 are disconnected from the pattern exhibited for the other years. Outliers can be identified for these mentioned years. These are the years with the most significant correlation values between the biomass and a vegetation index, as well as being the ones closest to zero, implying that no association exists. A color palette was added to the matrices to enable better visualization.

| **2013** | Biomass | MSAVI | CVI | GLI | TVIb |
|---|---|---|---|---|---|
| Biomass | 1.000 | 0.567 | 0.667 | 0.746 | 0.415 |
| MSAVI | 0.567 | 1.000 | 0.727 | 0.693 | 0.936 |
| CVI | 0.667 | 0.727 | 1.000 | 0.364 | 0.580 |
| GLI | 0.746 | 0.693 | 0.364 | 1.000 | 0.736 |
| TVIb | 0.415 | 0.936 | 0.580 | 0.736 | 1.000 |
| Number of Landsat images: 32 | | | | | |

| **2014** | Biomass | MSAVI | CVI | GLI | TVIb |
|---|---|---|---|---|---|
| Biomass | 1.000 | 0.763 | 0.761 | 0.799 | 0.640 |
| MSAVI | 0.763 | 1.000 | 0.795 | 0.845 | 0.959 |
| CVI | 0.761 | 0.795 | 1.000 | 0.526 | 0.672 |
| GLI | 0.799 | 0.845 | 0.526 | 1.000 | 0.869 |
| TVIb | 0.640 | 0.959 | 0.672 | 0.869 | 1.000 |
| Number of Landsat images: 48 | | | | | |

| **2015** | Biomass | MSAVI | CVI | GLI | TVIb |
|---|---|---|---|---|---|
| Biomass | 1.000 | 0.635 | 0.691 | 0.702 | 0.440 |
| MSAVI | 0.635 | 1.000 | 0.874 | 0.665 | 0.916 |
| CVI | 0.691 | 0.874 | 1.000 | 0.444 | 0.712 |
| GLI | 0.702 | 0.665 | 0.444 | 1.000 | 0.747 |
| TVIb | 0.440 | 0.916 | 0.712 | 0.747 | 1.000 |
| Number of Landsat images: 37 | | | | | |

| **2016** | Biomass | MSAVI | CVI | GLI | TVIb |
|---|---|---|---|---|---|
| Biomass | 1.000 | 0.680 | 0.657 | 0.758 | 0.543 |
| MSAVI | 0.680 | 1.000 | 0.841 | 0.757 | 0.948 |
| CVI | 0.657 | 0.841 | 1.000 | 0.445 | 0.709 |
| GLI | 0.758 | 0.757 | 0.445 | 1.000 | 0.811 |
| TVIb | 0.543 | 0.948 | 0.709 | 0.811 | 1.000 |
| Number of Landsat images: 49 | | | | | |

| **2017** | Biomass | MSAVI | CVI | GLI | TVIb |
|---|---|---|---|---|---|
| Biomass | 1.000 | 0.605 | 0.638 | 0.659 | 0.435 |
| MSAVI | 0.605 | 1.000 | 0.811 | 0.725 | 0.945 |
| CVI | 0.638 | 0.811 | 1.000 | 0.359 | 0.665 |
| GLI | 0.659 | 0.725 | 0.359 | 1.000 | 0.785 |
| TVIb | 0.435 | 0.945 | 0.665 | 0.785 | 1.000 |
| Number of Landsat images: 48 | | | | | |

| **2018** | Biomass | MSAVI | CVI | GLI | TVIb |
|---|---|---|---|---|---|
| Biomass | 1.000 | 0.061 | 0.999 | 0.061 | 0.051 |
| MSAVI | 0.061 | 1.000 | 0.037 | 0.834 | 0.958 |
| CVI | 0.999 | 0.037 | 1.000 | 0.020 | 0.032 |
| GLI | 0.061 | 0.834 | 0.020 | 1.000 | 0.851 |
| TVIb | 0.051 | 0.958 | 0.032 | 0.851 | 1.000 |
| Number of Landsat images: 47 | | | | | |

| **2019** | Biomass | MSAVI | CVI | GLI | TVIb |
|---|---|---|---|---|---|
| Biomass | 1.000 | 0.637 | 0.675 | 0.816 | 0.491 |
| MSAVI | 0.637 | 1.000 | 0.808 | 0.818 | 0.954 |
| CVI | 0.675 | 0.808 | 1.000 | 0.569 | 0.682 |
| GLI | 0.816 | 0.818 | 0.569 | 1.000 | 0.819 |
| TVIb | 0.491 | 0.954 | 0.682 | 0.819 | 1.000 |
| Number of Landsat images: 43 | | | | | |

| **2020** | Biomass | MSAVI | CVI | GLI | TVIb |
|---|---|---|---|---|---|
| Biomass | 1.000 | 0.789 | 0.776 | 0.863 | 0.690 |
| MSAVI | 0.789 | 1.000 | 0.806 | 0.871 | 0.969 |
| CVI | 0.776 | 0.806 | 1.000 | 0.579 | 0.683 |
| GLI | 0.863 | 0.871 | 0.579 | 1.000 | 0.883 |
| TVIb | 0.690 | 0.969 | 0.683 | 0.883 | 1.000 |
| Number of Landsat images: 35 | | | | | |

| **2021** | Biomass | MSAVI | CVI | GLI | TVIb |
|---|---|---|---|---|---|
| Biomass | 1.000 | 0.647 | 0.647 | 0.806 | 0.511 |
| MSAVI | 0.647 | 1.000 | 0.737 | 0.830 | 0.962 |
| CVI | 0.647 | 0.737 | 1.000 | 0.454 | 0.595 |
| GLI | 0.806 | 0.830 | 0.454 | 1.000 | 0.833 |
| TVIb | 0.511 | 0.962 | 0.595 | 0.833 | 1.000 |
| Number of Landsat images: 31 | | | | | |

| **2022** | Biomass | MSAVI | CVI | GLI | TVIb |
|---|---|---|---|---|---|
| Biomass | 1.000 | 0.152 | 0.988 | 0.163 | 0.121 |
| MSAVI | 0.152 | 1.000 | 0.104 | 0.831 | 0.953 |
| CVI | 0.988 | 0.104 | 1.000 | 0.060 | 0.085 |
| GLI | 0.163 | 0.831 | 0.060 | 1.000 | 0.855 |
| TVIb | 0.121 | 0.953 | 0.085 | 0.855 | 1.000 |
| Number of Landsat images: 32 | | | | | |

**Figure 6.** Correlation matrix between estimated biomass and vegetation indices.

### 3.3. Minimum, Maximum, and Average Values for the Estimated AGB in Each LULC Class

A zonal statistics algorithm established the primarily statistical functions of minimum, maximum, and average values between the estimated AGB and the distinct LULC classes. Table 2 and Figure 7 show the minimum values of estimated AGB (Mg.ha$^{-1}$) per year for each LULC class as returned by the zonal statistics algorithm. Pasture class showed a lower value for the AGB, while Forest plantation showed a higher value.

**Table 2.** The zonal statistics algorithm returns the minimum values of estimated AGB (Mg.ha$^{-1}$) per year for each LULC class.

| Class/Year | 2013 | 2014 | 2015 | 2016 | 2017 | 2018 | 2019 | 2020 | 2021 | 2022 |
|---|---|---|---|---|---|---|---|---|---|---|
| Forest Plantation | 17.125 | 19.163 | 16.475 | 19.397 | 14.481 | 16.731 | 6.160 | 3.549 | 12.814 | 8.763 |
| Pasture | 0.491 | 0.072 | 0.001 | 0.059 | 0.007 | 0.109 | 0.131 | 0.235 | 0.015 | 0.581 |
| Sugarcane | 10.304 | 11.002 | 11.802 | 9.538 | 10.332 | 11.884 | 10.126 | 11.700 | 11.430 | 10.770 |
| Mosaic of Uses | 0.006 | 1.054 | 5.270 | 5.828 | 3.522 | 1.354 | 4.714 | 4.478 | 1.885 | 0.679 |
| Soybean | 8.731 | 7.168 | 4.663 | 6.384 | 3.973 | 6.510 | 4.130 | 5.070 | 6.877 | 4.456 |
| Other Temporary Crops | 4.428 | 4.842 | 7.678 | 6.384 | 5.066 | 6.485 | 5.330 | 6.500 | 5.361 | 8.337 |

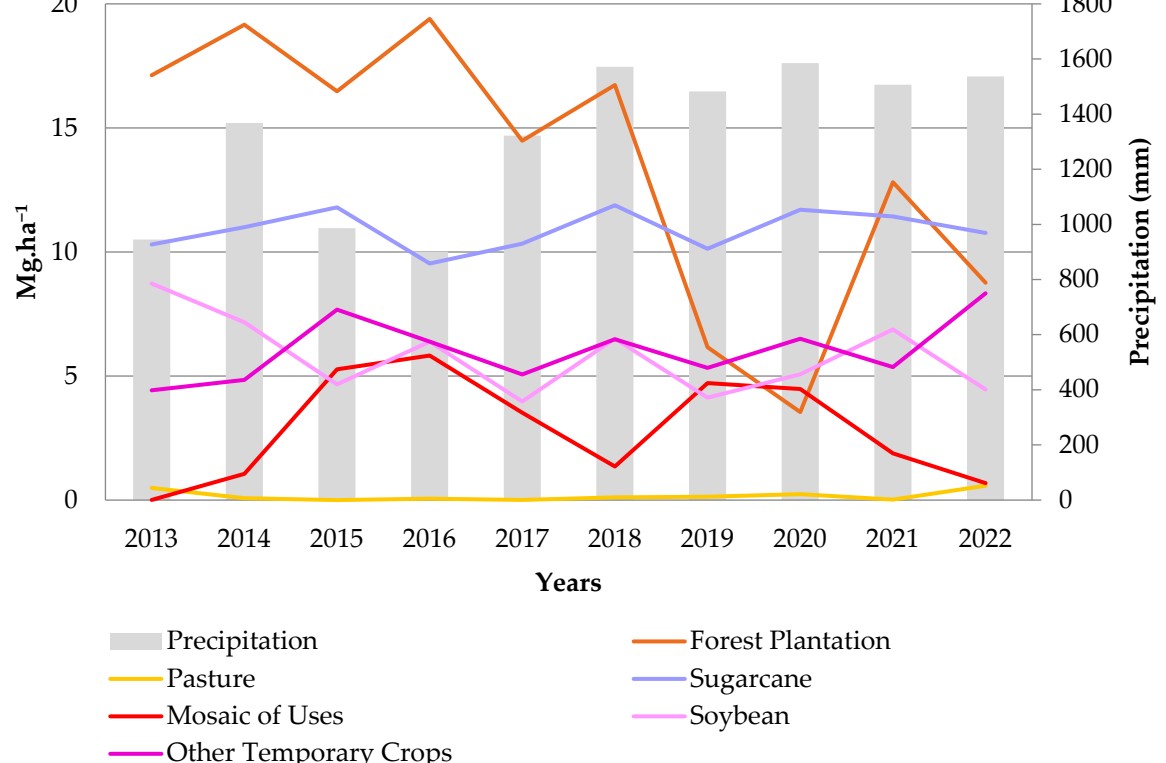

**Figure 7.** Line chart with the minimum values of estimated AGB (Mg.ha$^{-1}$) per year for each LULC class as returned by the zonal statistics algorithm.

Sugarcane exhibited elevated minimum values compared with the other classes, except for the Forest plantation category, which behaved as the patterns showed in precipitation. The minimum values for the Mosaic of uses, Soybean, and Other temporary crops classes are closely aligned. Nevertheless, abrupt transitions from one year to the next are prominent. As previously noted, the Pasture class demonstrates lower values, while the Forest plantation experienced a noteworthy decline between 2018 and 2020.

Table 3 and Figure 8 show the maximum values of estimated AGB (Mg.ha$^{-1}$) per year for each LULC class as returned by the zonal statistics algorithm. The most significant results are in the Pasture class, being 418.621 Mg.ha$^{-1}$ in 2013, which may suggest that forest areas were changed to pasture during the period. The lowest value was in 2015 in the Sugarcane class, with 53.972 Mg.ha$^{-1}$. Values for Soybean, Mosaic of uses, and Other temporary crops were relatively similar.

**Table 3.** The zonal statistics algorithm returns the maximum values of estimated AGB (Mg.ha$^{-1}$) per year and for each LULC class.

| Class/Year | 2013 | 2014 | 2015 | 2016 | 2017 | 2018 | 2019 | 2020 | 2021 | 2022 |
|---|---|---|---|---|---|---|---|---|---|---|
| Forest Plantation | 80.870 | 90.266 | 80.605 | 88.880 | 80.502 | 93.857 | 103.387 | 105.585 | 111.828 | 101.552 |
| Pasture | 418.621 | 123.609 | 107.757 | 108.187 | 115.871 | 109.411 | 217.967 | 130.737 | 197.711 | 122.180 |
| Sugarcane | 62.986 | 72.913 | 53.972 | 73.188 | 69.977 | 70.006 | 101.807 | 74.485 | 79.028 | 67.918 |
| Mosaic of Uses | 79.647 | 96.987 | 72.967 | 123.175 | 91.711 | 79.508 | 88.819 | 85.507 | 104.905 | 84.941 |
| Soybean | 88.692 | 85.940 | 121.060 | 89.856 | 115.517 | 103.836 | 126.882 | 113.292 | 114.313 | 87.608 |
| Other Temporary Crops | 77.229 | 87.137 | 119.012 | 109.988 | 95.616 | 83.531 | 95.094 | 90.283 | 96.127 | 76.388 |

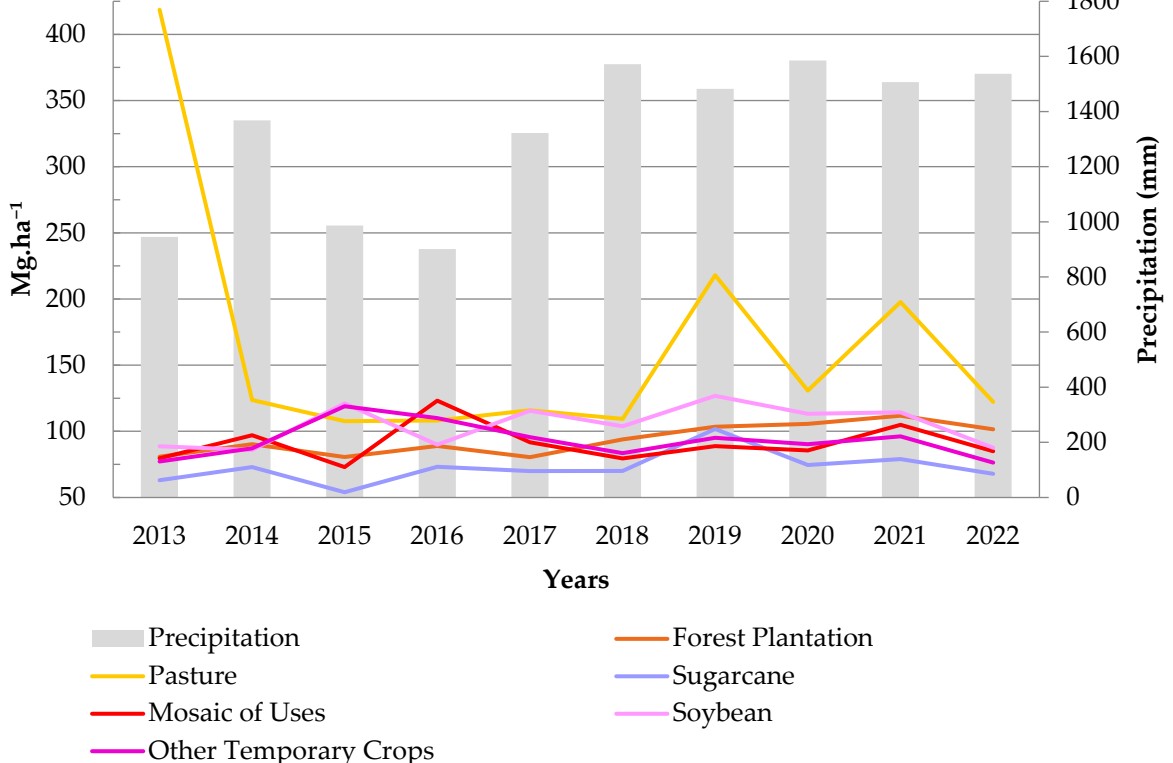

**Figure 8.** Line chart with the maximum values of estimated AGB (Mg.ha$^{-1}$) per year for each LULC class as returned by the zonal statistics algorithm.

The average values of estimated AGB (Mg.ha$^{-1}$) per year were returned from the zonal statistics. Table 4 and Figure 9 show the average values of estimated AGB (Mg.ha$^{-1}$) per year for each LULC class as returned by the zonal statistics algorithm. Over the ten years, the mean standard deviations (Mg.ha$^{-1}$) of the estimated AGB for each class were as follows: Forest plantation = 12.949; Pasture = 7.059; Sugarcane = 10.071; Mosaic of uses = 3.859; Soybean = 7.056; and Other temporary crops = 9.625. The Forest plantation class had the most significant standard deviation in 2020, with a value of 22.217 Mg.ha$^{-1}$, while the Mosaic of uses class had the lowest value of 2.966 Mg.ha$^{-1}$ in 2015.

The average values exhibited patterns of increasing and decreasing trends between 2013 and 2022. The categories encompassing Sugarcane and Other temporary crops demonstrated an increase in average values. On the other hand, the categories of Soybean and Mosaic of uses experienced a notable decrease in the average values over the period.

The precipitation data in Figures 7–9 enabled the observation of the dynamics over the LULC classes in agriculture–pasture mosaics. There was a significant rise in precipitation between 2013 and 2014, coinciding with an increase in the average AGB values for the Forest plantation, Pasture, and Other temporary crops classes in 2014. By relating the averaged AGB values (Figure 9) with the annual land surface temperature patterns (Figure 5), we

showed that 2015 had a high increase in average land surface temperature and a decreasing trend in precipitation.

**Table 4.** The zonal statistics algorithm returns the average values of estimated AGB (Mg.ha$^{-1}$) per year for each LULC class.

| Class/Year | 2013 | 2014 | 2015 | 2016 | 2017 | 2018 | 2019 | 2020 | 2021 | 2022 |
|---|---|---|---|---|---|---|---|---|---|---|
| Forest Plantation | 42.151 | 49.204 | 42.875 | 42.786 | 37.777 | 56.265 | 46.375 | 56.197 | 44.070 | 43.425 |
| Pasture | 22.563 | 25.353 | 19.220 | 23.078 | 22.672 | 23.421 | 20.956 | 25.440 | 25.088 | 22.855 |
| Sugar Cane | 26.268 | 24.631 | 20.342 | 31.470 | 26.333 | 33.537 | 25.143 | 33.347 | 35.475 | 38.189 |
| Mosaic of Uses | 24.196 | 25.087 | 22.144 | 23.699 | 23.642 | 23.671 | 23.168 | 23.900 | 24.881 | 22.936 |
| Soybean | 30.173 | 30.240 | 27.573 | 27.838 | 25.882 | 26.552 | 25.823 | 26.535 | 26.059 | 24.629 |
| Other Temporary Crops | 23.889 | 28.120 | 21.603 | 22.870 | 24.612 | 26.225 | 29.245 | 34.113 | 32.955 | 30.903 |

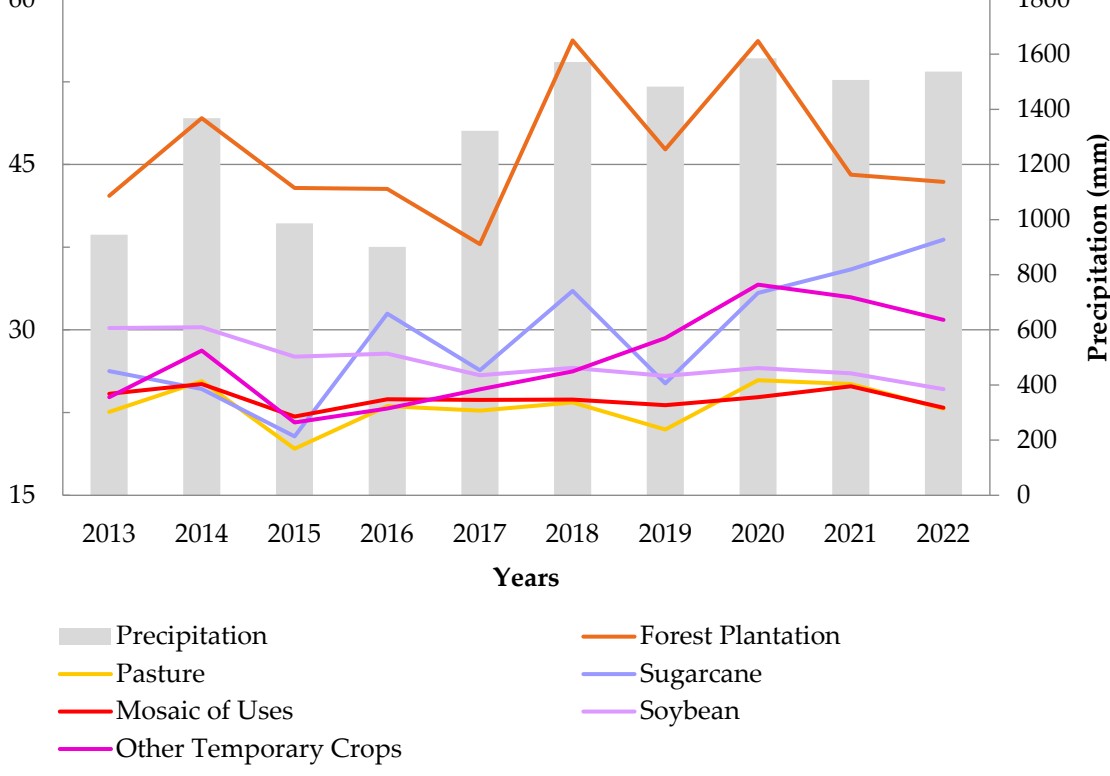

**Figure 9.** Line chart with the average values of estimated AGB (Mg.ha$^{-1}$) per year for each LULC class as returned by the zonal statistics algorithm.

Figure 10 presents the sum values of estimated AGB (Mg.ha$^{-1}$) per year for each LULC class as returned by the zonal statistics algorithm. The most extensive stocks of biomass are found in the Pasture, Mosaic of uses, Soybean, and Forest plantation classes. This pattern can be explained by the fact that these LULC categories have the most significant areas or types of cultivation that favor the most considerable amount of carbon stock in the territory of the Campo Maior Complex.

Table 5 shows the minimum, maximum, and average values of estimated AGB (Mg.ha$^{-1}$) for Forest formation. These values were considered for validation only. The lowest minimum values occurred in 2014, the average in 2015, and the maximum in 2018. Additionally, the mean standard deviation value for the Forest formation was 12.076 Mg.ha$^{-1}$. Figure 11 presents the data in a line graph to improve the visualization of the results for Forest formation.

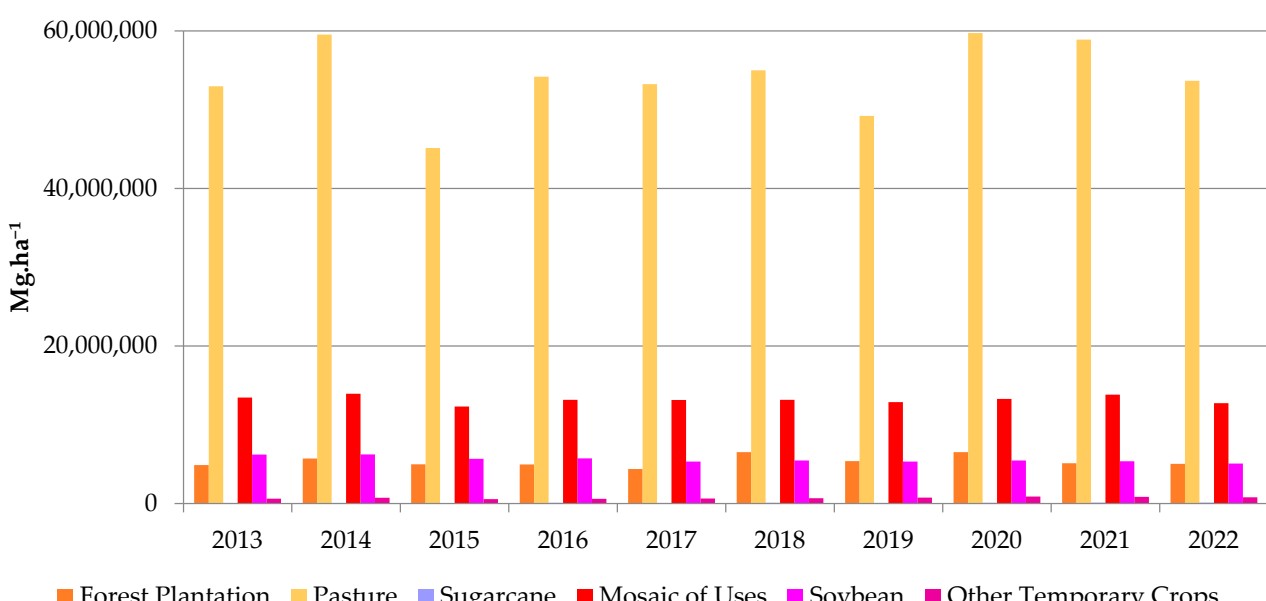

**Figure 10.** Sum values of estimated AGB (Mg.ha$^{-1}$) per year for each LULC class as returned by the zonal statistics algorithm.

**Table 5.** The zonal statistics algorithm returns the minimum, maximum, and average values of estimated AGB (Mg.ha$^{-1}$) per year for Forest formation.

| Forest Formation/Year | 2013 | 2014 | 2015 | 2016 | 2017 | 2018 | 2019 | 2020 | 2021 | 2022 |
|---|---|---|---|---|---|---|---|---|---|---|
| Minimum | 1.234 | 0.143 | 0.725 | 0.375 | 0.308 | 0.527 | 0.451 | 0.874 | 0.301 | 0.634 |
| Maximum | 137.085 | 127.000 | 121.100 | 122.795 | 116.780 | 114.881 | 276.963 | 169.960 | 203.311 | 123.300 |
| Average | 50.568 | 60.836 | 41.520 | 52.570 | 53.630 | 59.149 | 56.893 | 64.472 | 63.975 | 60.771 |

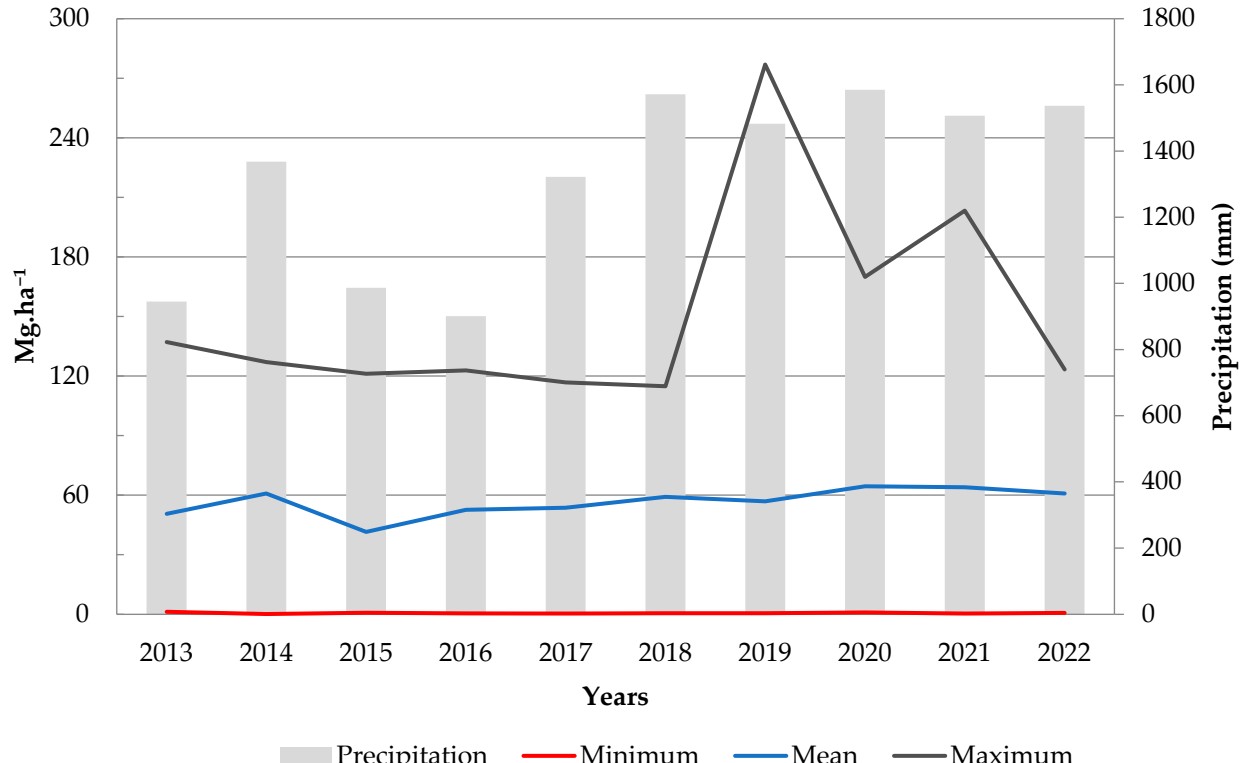

**Figure 11.** Line chart with the minimum, maximum, and average values of the estimated AGB (Mg.ha$^{-1}$) per year for Forest formation.

When examining the trend lines for each value, it is evident that there is an increase in the maximum values between 2019 and 2021. However, this trend is not reflected in the average values, as they appear to alternate between years of increase and decrease in the AGB dynamics. The most significant variation in average AGB values within the Forest formation class (Figure 11) occurred in 2015. It corresponds to the year with the highest annual land surface temperature and the most significant annual precipitation reduction (Figure 5).

## 4. Discussion

### 4.1. Biomass Estimation

It is highly recommended that agricultural practices be better managed, as they represent the primary vector of environmental degradation of the CMC native vegetation [30,50]. Due to its moderate climate risk, this region is experiencing an increasing agribusiness expansion into areas previously unattractive for such activities [30]. However, out of the 8,896 deforestation alerts recorded in Piauí, approximately 17% were identified within the boundaries of the CMC [50]. These activities must be compatible with the SI sustainability policy to minimize environmental and population consequences [38,39]. The results presented in Section 3.1 show changes in the estimated AGB values for the study area from 2013 to 2022, implying that land use changes are prominent in this area, resulting in the conversion of natural vegetation into crops.

In addition to the challenges of cost, periodicity, and compatible details of measuring aboveground biomass (AGB) in semiarid regions [16–18], factors such as soil type and moisture significantly influence vegetation indices (VIs) [21]. It is worth noting that NDVI is not recommended for modeling AGB values [24]. Therefore, by utilizing the estimator proposed in a similar study [10], it was possible to model and estimate AGB values for the agriculture–pasture mosaics in the CMC region.

Factors such as irrigation, fertilization, and management practices influence the AGB [57,58], and the modeling conducted here does not differentiate between these influences as it did not analyze the impact of these variables. However, Figure 5 provides information on land surface temperature and precipitation that aids in comprehending the potential alterations to the estimated AGB. It is particularly relevant due to the pronounced sensitivity of the CMC vegetation to water availability [45]. The uncertainties linked to AGB estimation through remote sensing [59,60] could potentially lead to underestimating high values and overestimating low values of the AGB [61,62].

The overestimated results, as indicated by the AGB values of 4 Kg.ha$^{-1}$ for Pasture areas in the Caatinga [63] shown in Figure 3, as well as the values extracted and presented in Tables 2–4 for the Pasture class, may be attributed to the accuracy of the agriculture–pasture mosaics extracted from MapBiomas. However, it is crucial to note that the CMC region also serves as an ecotone between the Caatinga and the savanna-like Cerrado biomes [30]. Furthermore, a temporal refinement was performed to identify the optimal period for each of the six classes within the agriculture–pasture mosaics to model the AGB values effectively.

In the southern portion of the CMC territory (Figure 3), it is observable that the Forest plantation class has exhibited progressive growth over the years, culminating in its highest aboveground biomass (AGB) values in 2021. The state of Piauí, in collaboration with the São Francisco and Parnaíba Valleys Development Company (CODEVASF), facilitated the establishment of the cellulose industry in the Baixo Parnaíba Valley. However, in 2014, the Suzano company withdrew from the project due to environmental lawsuits [64]. Nonetheless, the cultivation of planted areas for charcoal production purposes persists [64], and the company still maintains an extensive cultivation area in the municipality of Passagem Franca, Piauí.

### 4.2. Correlation Matrix

Most biomass stock studies using multispectral sensors reported in the literature use the Normalized Difference Vegetation Index (NDVI) and sensors with a spatial resolution higher than 30 m [65–67]. Another method involves calculating the $CO_2$ flux index, which is derived

by multiplying the NDVI by the rescaled positive values of the Photochemical Reflectance Index (PRI) [68,69]. However, it is still necessary to establish an imagery classification for qualitative thematic classes. Alternatively, Sentinel-2 data collection can be used [70]. However, in this study, Landsat 8 OLI enabled a larger space–time scale. In addition, specific MVI and VI reference values from Landsat 8 OLI for semiarid regions were used.

Figure 6 demonstrates the correlation matrix between the biomass results and the vegetation indices employed in the study (MSAVI, CVI, GLI, and TVI) as the best predictors for estimating biomass in agriculture–pasture areas [10] across semiarid regions. When evaluating the performance and correlation of the indices, it was usually noted that biomass and GLI had a higher correlation, while TVI had the lowest correlation but with highly representative results. We noticed a significant association between MSAVI and TVI among the VIs.

Investigating explanations for the poor statistical values in 2018 and 2022 is vital. Figure 5 demonstrates that 2018 and 2022 correspond to the second and third years of the period with the highest average annual precipitation. As discussed earlier, the CMC vegetation is responsive and acclimates to the dynamics of water availability [45]. Notably, the average land surface temperature exhibited an approximate 2 °C variation between 2015 and 2018 and between 2018 and 2022. This land surface temperature variation could potentially account for the observed low correlation values during these specific years.

### 4.3. Statistics by Each LULC Class

Other studies showed that semiarid regions present representative values of AGB, as in the case of agriculture–pasture mosaics [7–9]. Monitoring the AGB variation in agriculture–pasture mosaics may indicate changes in vegetation cover, loss of biodiversity, or soil degradation. The tables and figures in Section 3.3 show how the values vary by each land cover class. Sugarcane is a semi-perennial crop [70] that must always have greater AGB minimum values (Figure 7 and Table 2) than other agriculture classes such as Soybean.

The CMC region has a significant amount of Forest plantations since it is a place that had received incentives for cellulose manufacturing [64]. This land cover is distinguished by tall eucalyptus plantations that differ significantly from the other agricultural crops in the MapBiomas classifications [46]. The decrease in average values may indicate the time of forest harvesting. There are studies and literature with the theme of biomass estimation in pasture areas, especially in semiarid regions, but they are few, and there may be an overestimation of the values [10,61,62]. Figure 10 shows that Pasture, Mosaic of uses, and Soybean areas exhibit the highest biomass accumulation within the CMC territory. Considering that natural pastures are prevalent in the CMC region, how the pastures grow can impact the biomass results. Public strategies such as the ABC Plan with SI and agroforestry models [38,39] are critical for developing low-carbon agriculture in pasture regions and other agricultural areas.

The transition areas between Cerrado and Caatinga, such as the CMC territory, encompass vegetation that is susceptible to rainfall patterns and drought [30,37]. When utilizing secondary data from MapBiomas to identify Forest formation classes, the accuracy for these biomes ranges from 76% to 77% [46]. Studies on land use and land cover (LULC) changes in the CMC region are scarce, with the most recent data only available up to 2020 [30]. Consequently, the minimum values presented in the Forest formation areas (Figure 11 and Table 5) and the maximum values in the Pasture (Figure 7 and Table 2) may fall into different land cover classes. The Nascimento's estimator [10] was also developed for pasture and agriculture–pasture mosaic areas, potentially resulting in underestimated AGB values for Forest formation.

## 5. Conclusions

This study aimed to quantify biomass in the agriculture–pasture mosaics within the Campo Maior Complex (CMC) in Brazil. Remote sensing vegetation indices were used to estimate the AGB, and biomass estimation maps were created for different years. These maps revealed changes in land cover classes and biomass values over the analyzed 10-year

period. The field validation process for a large region requires many resources. However, the results obtained here from a previously determined estimator have proved satisfactory for a large area such as the CMC territory.

A correlation analysis was performed between the estimated biomass values and the vegetation indices used. The correlation matrix shows the relationship between the variables, emphasizing 2018 and 2022, given that they present values close to zero, without correlation between VIs and AGB values. This analysis helps to understand how different VIs adapted to the semiarid region relate to the estimated AGB values. Generating correlation matrices can assist future endeavors in developing biomass estimation equations tailored to specific land use and land cover classes. This approach stems from a retrospective analysis conducted using the data acquired in this study.

Agriculture, particularly commodity-focused activities, significantly contribute to greenhouse gas emissions. Agriculture must be considered more sustainably and take advantage of all the products of its production process as proposed by the Integrated Systems (SI). This approach promotes soil fertility, organic matter, and biomass production.

The information derived from modeling to estimate aboveground biomass (AGB) values benefits rural landowners and government management. For rural landowners, it aids in agricultural management, irrigation, and fertilization practices. Additionally, it facilitates the assessment of agricultural productivity. Government management can utilize AGB modeling to evaluate the environmental impact of farming and pasture practices. Variations in AGB values can indicate changes in land use and land cover, loss of biodiversity, and soil degradation. This information enables the formulation of conservation strategies, the identification of priority areas for environmental restoration, and the optimization of resource utilization.

**Author Contributions:** Conceptualization, Vicente de Paula Sousa Júnior, Javier Sparacino and Giovana Mira de Espindola; methodology, Vicente de Paula Sousa Júnior, Javier Sparacino, Giovana Mira de Espindola and Raimundo Jucier Sousa de Assis; software, Vicente de Paula Sousa Júnior; validation, Vicente de Paula Sousa Júnior and Javier Sparacino; writing—original draft preparation, Vicente de Paula Sousa Júnior; writing—review and editing, Giovana Mira de Espindola, Javier Sparacino and Raimundo Jucier Sousa de Assis; visualization, Vicente de Paula Sousa Júnior; supervision, Giovana Mira de Espindola and Raimundo Jucier Sousa de Assis. All authors developed and discussed the manuscript together and wrote the final paper. All authors have read and agreed to the published version of the manuscript.

**Funding:** This research was partially funded by the Brazilian National Council for Scientific and Technological Development (CNPq)—Grant Number 441950/2018-3 and Coordination for the Improvement of Higher Education Personnel—Process Number 88887.820341/2023-00.

**Data Availability Statement:** Not applicable.

**Acknowledgments:** The authors would like to thank the Federal University of Piauí (UFPI) for supporting this research.

**Conflicts of Interest:** The authors declare no conflict of interest. The funder had no role in the design of the study, in the collection, analysis, or interpretation; in the writing of the manuscript; or in the decision to publish the results.

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
