# Peer review of "Carbon Biomass Estimation Using Vegetation Indices in Agriculture–Pasture Mosaics in the Brazilian Caatinga Dry Tropical Forest"

_ijgi, doi:10.3390/ijgi12090354_

Round 1
Reviewer 1 Report
A bold attempt to further extend the application of spatial modelling in the field of biology/ecology by combining cloud computing, digital image processing and vegetation indices. Although, it is by no means novel research because there are several examples of integrated model assessments for Aboveground Biomass. For e.g., Pham et al 2019 https://doi.org/10.1016/j.rsase.2019.03.003). Authors have proposed innovation to include application in a highly managed landscape while accounting for seasonal differences which is commendable. An aspect that seems flawed is the accuracy and robustness of the random sampling strategy. For an effective assessment of AGB between class, the sampling approach should be revisited to include strata between land use. Without an adequate sampling strategy, it is hard to make an informed decision on the validity of the calculated statistics, trend analysis and how both truly represent the land uses within the study area. Specific comments/questions are provided in the succeeding paragraphs.
Furthermore, did the authors consider the potential of pixel spectral unmixing for this study? This is a technique much suitable for separating live, senescent vegetation and non-vegetated areas in a pixel. One will expect that pixels with 55% an 85% soybean cultivated area will equally have different spectral reflectance, which in turn affects the AGB values.
Pg 2 L48 – 60: There is a lack of details and explanation about the potential strengths and weakness of MVIs compared with traditional NDVIs. Authors have reported criticisms from past studies without any analysis of the methods and its potentials for agriculture-pasture mosaics and other managed landscapes.
Pg5 L194 – 197: How representative are these random points of crop types. Authors state the inclusion of “well-defined areas of forest plantation, soybean, sugar cane and other temporary crops”. Granted forest is a single class for sampling but it appears others land uses are lumped together.
Pg7 L221: What do you mean by “radically different”? I will recommend that authors use scientific phrase(s) when comparing quantities in academic writing.
Pg7 L236: I thought forest formations/plantations were distinct from other crop types in the analysis.
Pg8 L256: Authors will need to adopt stratified random sampling and perhaps controlled for class size to tackle lack of representation in different land use categories.
P13, Figure 9: why is theere emphasis on showing the linear trend for forest formation only? The title and preliminary pages of the paper suggests a focus on agriculture-pasture mosaics.
Discussion and conclusion: More context should be given to the presented results. To begin, AGB is also affected by inputs (e.g., fertilisation and irrigation), environmental factors and management practices. Therefore, it is prudent to state that the modelling approach is unable to account for differences resulting from farm inputs and other environmental factors. See the following papers for evidence of inputs and the environment on AGB Dieleman et al (2012) Glob Change Biol, 18: 2681-2693. https://doi.org/10.1111/j.1365-2486.2012.02745.x and Garten Jr et al 2011. Biomass and Bioenergy, 35(1), pp.214-226. https://doi.org/10.1111/j.1365-2486.2012.02745.x
There are no major typographical errors, but authors should aim to improve the grammer syntax in several places.
Reviewer 2 Report
Check the correct spelling throughout the text and figures: “mg.ha-1” or “Mg.ha-1”.
Fig. 3, line 210: remove extra dot.
Fig. 4: missing symbol “с”.
Fig. 6 is not mentioned in the text.
Line 271: change table 3 to table 4.
Line 337: “To the south of the CMC (Figure 3), it is observable that the Forest plantation” - in Figure 4, mark where the fragments are located relative to the entire area.
Why there is an increase or decrease in indicators - it would be interesting to see the correlations of indices and biomass with temperature and precipitation.
Reviewer 3 Report
This study quantifies AGB in agriculture-pasture mosaics within Brazil's Campo Maior Complex. The methodology employs remote sensing cloud processing and utilizes an estimator incorporating the vegetation indices. The results of this study may contribute to provide a reference for managing the territory and the environment in a semiarid context. However, there are some concerns that the authors should address before it can be considered for publication.
(1) I suggest the authors highlight the research significance of this article in the last paragraph of the introduction.
(2) In the introduction, I suggest the authors add some recent references to back "China is a sensitive region affected by climate change (e.g., Zheng et al, 2021; Jiang et al., 2022)".
(3) Line 152-153, what is the basis for selecting "different thresholds" in the study?
(4) In the data, I suggest the authors add more information about data, such as data availability and access.
(5) More mechanism explanations should be added to further explain the influence of driving factors on accumulated temperature.
(6) In order to further highlight the innovation of this article, it is better to compare the results of this study with other studies.
(7) A paragraph of limitation discussion should be added to clarify the limitation or uncertainty of data and methods in this current study. For example, the uncertainty of remote sensing data (Li et al., 2022; Shen et al., 2022) may affect the research results.
(8) The conclusion is not a simple restatement of the results. The authors should further clarify the contribution of the research results to the research field.
References:
Impacts of climate change and anthropogenic activities on vegetation change: Evidence from typical areas in China. Ecological Indicators, 2021, 126: 107648.
Interannual variability of vegetation sensitivity to climate in China. Journal of Environmental Management, 2022, 301: 113768.
Asymmetric impacts of diurnal warming on vegetation carbon sequestration of marshes in the Qinghai Tibet Plateau. Global Biogeochemical Cycles, 2022, 36: e2022GB007396.
Uncertainty of city-based urban heat island intensity across 1112 global cities: Background reference and cloud coverage. Remote Sensing of Environment, 2022, 271: 112898.
Round 2
Reviewer 1 Report
Revision is satisfactory.
Reviewer 3 Report
The authors have addressed all my concerns. I suggest accept this paper in its present form.